# Anatomical Registration of Implanted Sensors Improves Accuracy of Trunk Tilt Estimates with a Networked Neuroprosthesis

**DOI:** 10.3390/s24123816

**Published:** 2024-06-13

**Authors:** Matthew W. Morrison, Michael E. Miller, Lisa M. Lombardo, Ronald J. Triolo, Musa L. Audu

**Affiliations:** 1Department of Biomedical Engineering, Case Western Reserve University, Cleveland, OH 44106, USA; ronald.triolo@case.edu (R.J.T.); mxa93@case.edu (M.L.A.); 2Motion Study Laboratory, Louis Stokes Cleveland Department of Veterans Affairs Medical Center, Cleveland, OH 44106, USA; michael.miller9@va.gov (M.E.M.); lisa.lombardo2@va.gov (L.M.L.)

**Keywords:** trunk control, sensor reorientation, networked neuroprosthesis, spinal cord injury, musculoskeletal

## Abstract

For individuals with spinal cord injuries (SCIs) above the midthoracic level, a common complication is the partial or complete loss of trunk stability in the seated position. Functional neuromuscular stimulation (FNS) can restore seated posture and other motor functions after paralysis by applying small electrical currents to the peripheral motor nerves. In particular, the Networked Neuroprosthesis (NNP) is a fully implanted, modular FNS system that is also capable of capturing information from embedded accelerometers for measuring trunk tilt for feedback control of stimulation. The NNP modules containing the accelerometers are located in the body based on surgical constraints. As such, their exact orientations are generally unknown and cannot be easily assessed. In this study, a method for estimating trunk tilt that employed the Gram–Schmidt method to reorient acceleration signals to the anatomical axes of the body was developed and deployed in individuals with SCI using the implanted NNP system. An anatomically realistic model of a human trunk and five accelerometer sensors was developed to verify the accuracy of the reorientation algorithm. Correlation coefficients and root mean square errors (RMSEs) were calculated to compare target trunk tilt estimates and tilt estimates derived from simulated accelerometer signals under a variety of conditions. Simulated trunk tilt estimates with correlation coefficients above 0.92 and RMSEs below 5° were achieved. The algorithm was then applied to accelerometer signals from implanted sensors installed in three NNP recipients. Error analysis was performed by comparing the correlation coefficients and RMSEs derived from trunk tilt estimates calculated from implanted sensor signals to those calculated via motion capture data, which served as the gold standard. NNP-derived trunk tilt estimates exhibited correlation coefficients between 0.80 and 0.95 and RMSEs below 13° for both pitch and roll in most cases. These findings suggest that the algorithm is effective at estimating trunk tilt with the implanted sensors of the NNP system, which implies that the method may be appropriate for extracting feedback signals for control systems for seated stability with NNP technology for individuals who have reduced control of their trunk due to paralysis.

## 1. Introduction

Individuals with spinal cord injuries (SCIs) often experience complete or partial loss of sensory and motor function at and below the level of their injury, which includes their extremities and trunk. This results in a loss of trunk stability and an impairment of their posture, making it difficult to complete many activities of daily living (ADLs) such as eating, dressing, and transferring [1,2]. As such, this population has consistently considered restoration of trunk stability to be a high research priority [3,4,5].

For several decades, functional neuromuscular stimulation (FNS) has been explored to address the immobility caused by paralysis due to SCI. FNS involves activating paralyzed muscles by applying small electrical currents to peripheral motor nerves via skin or implanted electrodes [6]. This practice has proven to be a promising tool for enabling individuals with SCI to perform many tasks that would otherwise be extremely difficult or impossible, including stepping [7,8], cycling [9], and bimanual reaching [10,11]. Some of these functional motions of the otherwise paralyzed limbs, such as cycling, were accomplished in a feedforward manner, in which the stimulation input parameters (pulse widths, amplitudes, and frequencies) were specified a priori and unresponsive to changes in the system or the environment. This type of control is convenient when motions are repetitive and not subject to changing external or internal factors but is inappropriate when disturbances that can affect the desired motion are applied to the system. For this reason, feedback control of FNS to compensate for such perturbations has become a prominent topic of interest in research regarding seated trunk stability [12].

An accurate measure of the current state of a system is necessary for the effective implementation of any feedback controller due to the inherent reliance of feedback control on the output of the control system. Inertial measurement units are electronic devices capable of measuring forces, accelerations, and orientations of rigid bodies. As such, they have become a powerful tool for providing such estimates for feedback control of seated trunk stability due to their small size and ability to output data that can be readily transformed into various relevant quantities, including tilt [13]. Many recent experiments have used a single accelerometer placed externally on the trunk to estimate tilt or orientation for this purpose [2,14,15,16]. To obtain accurate measurements using this method, the sensor had to be carefully and securely located such that its axes remained aligned with the anatomical axes of the trunk. This approach generally makes it difficult to use accelerometers for feedback control outside of the laboratory setting due to the need for consistency of the sensor placement and repeated donning and doffing of sensors.

The Networked Neuroprosthesis (NNP) developed at Case Western Reserve University in Cleveland, OH offers a novel and innovative way to overcome these limitations [17,18]. The NNP is a scalable, modular, fully implanted FNS system consisting of a central “power module” and a network of serially linked “remote modules” capable of recording biopotentials (i.e., electromyographic signals) or delivering stimulation. The power module powers the remote modules and is responsible for internal and external system communication. In addition, each remote module contains a three-axis accelerometer from which segmental kinematic variables, such as trunk tilt, can be estimated.

The development of the NNP provides an alternative solution to the problems faced in previous systems requiring externally placed sensors and makes a home-going feedback system more realistic. One major limitation of implanted sensor systems, however, is that the orientations of the modules are generally unknown after being installed. They are located due to surgical considerations that minimize the stresses placed on the devices and surrounding soft tissues, are not affixed to the bony skeleton, and are subject to time-dependent influences such as natural encapsulation processes. Previous studies performed for tracking human movements using accelerometers suggest that reorienting sensor signals can improve the accuracy of movement estimates [19,20,21]. Thus, to obtain accurate, robust measurements of trunk tilt, the orientation of the sensor outputs must first be determined and aligned with the anatomical axes of the torso of the recipient.

Imaging techniques such as CT scanning could be explored to obtain estimates of sensor orientation with respect to the body. However, this would require several cross-sectional images from various angles, requiring radiation exposure and perhaps complex image processing. Also, the modules are not guaranteed to remain in the same position over time. As such, these techniques may require repeated imaging sessions to reassess the orientation of the modules relative to the anatomy. The development of a simple and efficient method of reorienting the sensor signals from the NNP is needed before the system can be considered practical for implementation in feedback control systems for stabilizing the trunk.

In this study, a method to align the signals from the implanted NNP sensors with the anatomical axes of the trunk was developed and justified in an in silico model by comparing tilt estimates derived from simulated, unrotated accelerometer outputs to tilt estimates derived from reoriented accelerometer outputs of the algorithm. This approach was then verified in human trials with recipients of the NNP. Both raw, unrotated accelerometer signals and the reoriented signals were transformed into measurements of trunk tilt, and the outputs from each data set were compared to the gold standard derived from motion capture data to assess the robustness of the method.

## 2. Materials and Methods

### 2.1. Reorientation and Tilt Estimation Methods

To align the accelerometer signals with the anatomical axes of the body, an algorithm developed by Jiang et al. [19] and Chen et al. [20] was applied. This algorithm was chosen due to its effectiveness regarding the reorientation of externally placed accelerometer signals for the purpose of human activity recognition. The algorithm employs the Gram–Schmidt method of orthonormalization to define transformation matrices to align the axes of each sensor with the anatomical axes of the body to which it is attached while in a particular reference configuration. A pseudocode describing the process for determining the transformation matrices for each sensor is shown in Algorithm 1. Further explanation of the Gram–Schmidt method can be found in Jiang et al. [19].
**Algorithm 1.** Determination of Transformation MatricesSet normalized, standard signal vectors for erect and supine postures: h_1_,_st_ = [0 0 1]^T^ and         h_2_,_st_ = [−1 0 0]^T^.Use h_1_,_st_ and h_2_,_st_ in the Gram-Schmidt orthonormalization process to compute three         orthonormal column vectors in the standard orientation: r_1_,_st_, r_2_,_st_, r_3_,_st_.Compute the standard direction cosine matrix: K_st_ = [r_1_,_st_ r_2_,_st_ r_3_,_st_].        **For** all active sensors                  Collect 3-axis accelerometer signal data from erect and supine postures                  Take the mean values of the outputs of each axis for both erect and supine trials.                  Calculate the raw signal vector related to the erect posture: h_1_,_raw_ = [x_erect_,_avg_
                          y_erect_,_avg_ z_erect_,_avg_]^T^.                  Calculate the raw signal vector related to the supine posture: h_2_,_raw_ = [x_supine_,_avg_
                          y_supine_,_avg_ z_supine_,_avg_]^T^.                  Use h_1_,_raw_ and h_2_,_raw_ in the Gram-Schmidt orthonormalization process to                           compute three orthonormal column vectors: r_1_,_raw_, r_2_,_raw_, r_3_,_raw_.                  Set up the raw direction cosine matrix: K_raw_ = [r_1_,_raw_ r_2_,_raw_ r_3_,_raw_]                  Calculate the rotation matrix between the standard and raw orientation: N =                           K_raw_ ⋅ K_st_^T^.        **End**

In this process, h_1_ and h_2_ are three-by-one, noncollinear signal vectors in three-dimensional, Euclidean space. The standard values for these vectors were chosen based on the assumption that the accelerometers were properly aligned with a rigid segment whose principal axes followed the standard aerospace convention. The raw values for these vectors were determined by taking the average values from preliminary trial data collected in the erect and supine postures. The rotation matrices found for each sensor using this algorithm were then applied to the sensor data collected for each subsequent trial. The new, reoriented signal outputs were used to estimate trunk tilt based on Equations (1) and (2) [14,22] for the pitch and roll values, respectively.
(1)$$Pitch=\tan^{-1}\left(\frac{-ac_{x}}{\sqrt{{ac_{y}}^{2}+{ac_{z}}^{2}}}\right)$$
(2)$$Roll=\tan^{-1}\left(\frac{ac_{y}}{ac_{z}}\right)$$

### 2.2. In-Silico Proof of Concept

#### 2.2.1. Model

To verify the accuracy and robustness of the sensor alignment method, an anatomically realistic musculoskeletal model of the trunk was developed using OpenSim 4.4 software (Simbios, Stanford, CA, USA). The skeletal mesh of the trunk model is shown in Figure 1. Five virtual, randomly oriented accelerometer sensors represented by small cuboids in the model were placed on the trunk. The locations of the virtual sensors were assigned based on the general locations anticipated for the implanted modules of the NNP as well as locations previously used for experiments with external sensors. The sensor axes were oriented at randomly generated, known angles with respect to the anatomical axes of the trunk.

We explored the sensitivity of the algorithm to better understand the potential performance under real-world conditions by introducing various distortions to the ideal signals produced by the virtual sensors. In this process, we examined the following signal conditions: no noise or other variations (an ideal signal), 10% random noise, 0.002 g/s drift, and 0.15 g offsets. The applied distortions were based on the characteristics listed in the documentation found in Appendix A regarding the actual accelerometers embedded in the NNP modules. Then, 18 s of static simulated data for both erect and supine postures were created for each type of distortion applied to the output from each axis of the virtual sensors. In addition, 18 s of simulated dynamic leaning data were recorded for each type of irregularity. During these dynamic trials, the model was moved from an erect seated position to positions of 30° of trunk flexion, 30° of right lateral flexion, 30° of trunk extension, and 30° of left lateral flexion with brief returns to the erect position between each posture.

#### 2.2.2. Data Analysis and Statistical Testing

The averages of the output signals from each axis at each simulated timepoint for the erect and supine posture trials were used as the raw accelerometer vectors in Algorithm 1 to determine the rotation matrices that reoriented the signals from the virtual sensors to coincide with the anatomical axes of a model trunk. The tilt estimates from simulated sensor outputs were compared to the tilt angle targets by determining the correlation coefficients (R values) and the root mean square errors (RMSEs) for the signals from individual model sensors. Tilt estimates made using Equations (1) and (2) were derived from both unrotated and reoriented accelerometer signals. These estimates were then used to assess the differences between the unrotated tilt outputs and the rotated outputs. The averages and standard deviations of the R and RMSE values across all conditions were computed for each sensor, and Wilcoxon Rank Sum nonparametric tests were performed to test for significant differences between the unrotated, simulated tilt estimates and those calculated under each distortion condition to assess the robustness of the method.

### 2.3. Experimental Verification

#### 2.3.1. Participants

Three individuals with SCI who received implanted NNP systems participated in this study. Table 1 shows the anthropometric and neurological injury data for each participant as well as the number of trunk modules containing embedded accelerometers. In addition, a layout of the modules for a typical NNP trunk system and their corresponding electrodes is shown in Figure 2. A typical configuration of NNP modules for trunk control would include three stimulating remote modules (1–3), and one remote module (9) for biopotential recording. Participants had varying numbers of trunk modules because the NNP system was customized for each implant recipient based on their individual needs. In the case of S1, normally four trunk modules—three four-channel stimulating and one two-channel biopotential recording module—would be present, but the biopotential recording module had to be explanted due to clinical issues unrelated to technical performance. In S3, only the biopotential recording module yielded signals indicative of trunk motion since the other modules for their system were located in the arm.

Participants were informed of all aspects of the experiment and signed consent forms approved by the local institutional review board (IRB: VA Northeast Ohio Healthcare System, Protocol Number: 1695449, Approval Date: 12/08/22).

#### 2.3.2. Experimental Setup

Signals were collected from the KXTE9-2050 2 g Tri-Axis Accelerometers (Kionix, Ithaca, NY, USA) embedded in each NNP trunk module. Output signals from the implanted sensors were collected at a rate of 50 Hz via a Speedgoat real-time computer (MathWorks, Natick, MA, USA) using a custom Simulink 10.2 model (MathWorks, Natick, MA, USA). While this study does not explore the use of stimulation via the NNP, the collection rate of 50 Hz was chosen to ensure that the future Simulink models that control stimulation parameters can collect multiple accelerometer readings between stimulation pulses, which occur at a frequency of 20 Hz. Although using this rate means that there will be some delay present in the trunk tilt estimates, it is negligible in this study due to its short duration compared to the slow, quasi-static leaning movements in question. It should also be noted that the proposed signal reorientation method was applied offline and as such had no effect on the real-time collection of the accelerometer outputs from the Speedgoat computer and Simulink model. The application of the resulting transformations is only a series of matrix operations that are extremely computationally efficient, requiring approximately 12.6 ms to be completed.

In addition, retroreflective markers were placed on the skin over the C7 vertebrae, the T10 vertebrae, the sacrum, the left clavicle, the sternum, bilaterally on the acromion of the scapula, the middle of the humerus over the lateral head of the triceps, the lateral epicondyle of the ulna, the anterior superior iliac spine, and the posterior inferior iliac spine. Trunk kinematics were computed from the positions of these markers collected by a 16-camera motion capture system (Vicon Motion Systems Ltd., Oxford, UK) at a frequency of 100 Hz.

#### 2.3.3. Experimental Procedure

Each experiment consisted of six static and 10 dynamic trials. All trials were performed with participants seated on a portable examination table with assistance from a physical therapist due to the lack of voluntary motion below the subject’s level of injury. For static trials, subjects were held in erect sitting, 30° of trunk flexion, 30° of trunk extension, 30° of right lateral flexion, 30° of left lateral flexion, and supine positions for approximately 10 s each. Estimates of these postures were ascertained with a handheld goniometer and were not strictly enforced since gold standard values were obtained from the motion capture system. The dynamic trials consisted of 10 multi-posture movements, which started with erect sitting, holding the position for five seconds, moving to a posture of trunk flexion and holding for approximately five seconds, returning to an erect posture, and repeating this sequence for right lateral flexion, extension, and left lateral flexion.

#### 2.3.4. Data Analysis and Statistical Testing

The reorientation algorithm was applied to the accelerometer signals collected from the experimental trials. The averages of the accelerometer signals from each axis for the erect and supine posture trials were used as the raw accelerometer vectors to determine the rotation matrices for each implanted sensor that aligned the sensor signals with the anatomical axes of the trunk. The reoriented signals were determined by applying the methods in Algorithm 1. Trunk tilt was derived by applying Equations (1) and (2) to all trials performed in a testing session. Tilt estimates were derived from raw and reoriented signals so that comparison could be performed to statistically assess the performance of the reorientation algorithm. These tilt estimates from both data sets were then compared to trunk tilt obtained from motion capture data.

The accuracy of the tilt estimates derived from the individual accelerometers relative to the gold standard motion capture data was determined from the R and RMSE values between data sets. For each sensor, the averages and standard deviations of these measures were assessed. Wilcoxon Rank Sum nonparametric tests were performed for each sensor to test for significant differences between the tilt estimates made using the raw sensor signals and those made using the reorientation algorithm.

## 3. Results

### 3.1. In Silico Proof of Concept

As expected, unrotated, simulated signals from the randomly oriented sensors all exhibited inaccuracies in comparison to the true orientation of the model trunk (Figure 3).

This comparison shows the necessity for reorienting sensor signals to obtain relevant estimates of trunk tilt.

For most conditions, the correlation coefficients were higher and RMSEs lower for tilt estimates from each modeled sensor under each type of signal variation after reorientation than from simulated, unrotated tilt estimates as shown in Figure 4. The unrotated signals were found to have R values with averages of 0.67 ± 0.32 and 0.70 ± 0.39 for pitch and roll, respectively, as well as RMSE values of 32.32 ± 12.07° and 27.14 ± 15.07° for pitch and roll, respectively. For all other conditions, R values were found with averages ranging from 0.96 ± 0.02 to 0.99 ± 0.01 and RMSE values ranging from 1.50 ± 1.27° to 3.55 ± 1.06° for all trunk tilt estimates under the various distortion conditions. All conditions except for the addition of 10% noise resulted in statistically significant improvement, with correlation coefficients significantly higher (*p* < 0.05) and RMSEs significantly lower (*p* < 0.01), after applying the transformation compared to the estimates made from the raw simulated signals. While the difference in the sensor-averaged correlation coefficients could not be considered significant in the case of 10% added noise, there was still a large increase in the average correlation coefficients for both pitch and roll estimates.

### 3.2. Experimental Verification

Examples of the tilt estimates derived from reoriented accelerometer signals collected during the multi-posture dynamic trials for each participant can be seen in Figure 5. In each case, estimates from all the sensors capable of recording acceleration during the experiment are shown. Overall, most of the modules for each subject seem to correlate well with the motion capture data, with the most obvious variations in tilt estimates occurring in module 2 of subject S2 and module 9 of subject S3.

Figure 6 shows the correlation coefficients and RMSE values calculated for each participant’s unrotated and reoriented sensor data by comparing tilt estimates from each group to the motion capture data. Regarding pitch tilt estimates in S1, the average R values for unrotated estimates ranged from −0.89 ± 0.02 to 0.91 ± 0.02 while reoriented estimates ranged from 0.88 ± 0.03 to 0.94 ± 0.02. In addition, the average unrotated RMSE values for pitch estimates ranged from 47.35 ± 0.78° to 61.22 ± 2.00° while reoriented RMSE values ranged from 11.05 ± 0.91° to 12.06 ± 0.63°. For the roll estimates in the same subject, the average R values for unrotated estimates ranged from −0.49 ± 0.15 to 0.93 ± 0.02 while reoriented estimates ranged from 0.86 ± 0.02 to 0.97 ± 0.01. The average unrotated RMSE values for roll estimates ranged from 69.66 ± 1.47° to 152.02 ± 1.78° while reoriented RMSE values ranged from 10.53 ± 0.72° to 11.39 ± 1.65°. In cases of both R and RMSE values for pitch and roll estimates, the variation across sensors decreased, and statistically significant differences were found between the unrotated and reoriented data sets.

In S2, the average R values for unrotated pitch estimates ranged from −0.67 ± 0.06 to 0.72 ± 0.04 while reoriented estimates ranged from ranged from 0.74 ± 0.08 to 0.94 ± 0.01. The average unrotated RMSE values for pitch estimates ranged from 55.84 ± 1.45° to 77.71 ± 1.33° while reoriented RMSE values ranged from 6.65 ± 0.64° to 13.04 ± 0.89°. For the roll estimates in the same subject, the average R values for unrotated estimates ranged from −0.75 ± 0.08 to 0.37 ± 0.05 while reoriented estimates ranged from 0.58 ± 0.07 to 0.95 ± 0.01. The average unrotated RMSE values for roll estimates ranged from 66.66 ± 2.36° to 155.21 ± 3.87° while reoriented RMSE values ranged from 8.56 ± 0.45° to 26.40 ± 4.77°. The variation across sensors again decreased, and statistically significant differences were found between the unrotated and reoriented data sets for all values except those regarding the correlation coefficients of the pitch tilt estimates.

In S3, the average R value for the unrotated pitch estimates was found to be 0.38 ± 0.05 and 0.78 ± 0.04 for the reoriented estimates. The average unrotated RMSE value for pitch estimates was 61.20 ± 1.02°, and the reoriented RMSE value was 12.52 ± 1.30°. For the roll estimates in the same subject, the average R value for the unrotated estimate was 0.62 ± 0.03 while the reoriented estimate average was 0.92 ± 0.02. The average unrotated RMSE value for roll estimates was 59.94 ± 0.97° while the reoriented RMSE value was 10.27 ± 0.83°. Similarly to S1, statistically significant differences were found between the unrotated and reoriented data sets for all.

## 4. Discussion

The outcomes from the in silico study indicate that the algorithm can accurately reorient the signals from tri-axis accelerometers to align with the anatomical axes of the trunk. In addition, they suggest that expected levels of noise or other distortions from the accelerometer sensors have little effect on the outcomes of the reorientation algorithm. The analytical values collected from these experiments are similar to those obtained in other studies that used externally placed accelerometers to estimate trunk tilt such as Wong and Wong [23] (which reported RMSEs below 5°) and Friederich et al. [22] (which found RMSEs below 8° and R values above 0.93).

The rotation of the implanted NNP sensor signals with the method presented in this work resulted in correlation values greater than or equal to 0.78 and RMSEs less than 13° for most cases in both pitch and roll angle estimates of trunk tilt. While these values are more variable than those obtained in other studies that used accelerometers to estimate trunk tilt [22,23], our results suggest that the proposed method for reorienting arbitrarily oriented sensors to be aligned with the anatomical axes of the trunk might be valuable for providing significantly more accurate data for estimating trunk tilt in a feedback control system than from unrotated accelerometer data. Prior studies differ from the current study in that they utilized sensors placed on the surface of the skin rather than being implanted. While implantation benefits the implant recipient by removing the requirement for donning and doffing external sensors, the signals from the implanted sensors will be affected by additional, random noise due to interference from the encapsulation after implantation as well as from other tissue that may be surrounding the sensors [24,25]. Future analyses could explore filtering to mitigate the effects of such noise and further enhance trunk tilt estimation.

There are limitations to this study that warrant further investigation of estimating trunk tilt using the NNP system. One has to do with the observation that sudden or unexpected changes in the tilt estimates were observed in certain leaning postures for some of the modules that were assessed. This phenomenon can be seen in Figure 5, specifically in the outputs collected from M2 in S2 when there is an unexpected increase in the roll value near the 10-s mark and M9 in S3 when the sign of the tilt output flips near the 60-s mark. This may be caused by singularities in the Euler angle computations, resulting in a sign flip in the output whenever the raw tilt estimates are close to 90 degrees. Another reason these sign flips might occur is due to the limitations of the tilt estimation method, further explanation of which can be found in Pedley [26], which explores the use of tri-axis accelerometers in cell phones. Problems like this could be mitigated by using sensor fusion algorithms that utilize signal contributions from all sensors. Examples of this practice that might be relevant for the NNP system include the work of Friederich et al. [22], which optimized weighted averages to determine the contributions from each sensor included in the algorithm, and the work of Lui and Liu [27], which employed a Naïve Bayesian classifier to recognize human activities from multiple wearable sensors. Another reason for these errors could be due to placement within the body. For example, M2 in subject S2 is located subcutaneously over the abdominal muscles. This location mostly consists of soft tissue and muscle that could allow the modules to change orientation during movement. Methods for relieving these types of errors are still under investigation.

The assumptions made regarding the accelerometer outputs could also have limited the study. That is: gravity is the only source of acceleration being applied, whereas the sensor signals reflect all types of acceleration of the body. While gravitational acceleration is the most prominent acceleration component in static postures or during slow movements when additional components of acceleration are near zero, this assumption becomes problematic when quick, abrupt changes in motion are made by the implant recipient that may cause sudden spikes in the translational, rotational, and/or Coriolis components of the acceleration vectors being measured. For use in a feedback control system for seated stability in individuals with highly impaired volitional trunk control, we do not expect this to be an issue due to the low-velocity nature of movements for leaning or other seated ADLs. In cases where quicker, more dynamic motions are expected, a method would need to be developed to isolate the gravitational acceleration for use as a feedback signal related to tilt.

The long-term stability of the implanted sensor signals was not specifically assessed throughout the time those data were collected during these experiments. However, any changes in the implanted sensor orientation over time were accounted for by performing the calibration process at the beginning of each experimental session.

## 5. Conclusions

An algorithm for estimating trunk tilt using the implanted accelerometer signals of the NNP was developed and tested in both an in silico study and in real-life implant recipients. The results from the reorientation of implanted sensor signals were similar to those obtained from a motion capture system as the gold standard and outperformed estimates of trunk tilt from raw, unrotated accelerations. The realignment process can be performed post-implantation and suggests that modules can be implanted due to surgical considerations without excessive regard for orientation with respect to the anatomical axes of the trunk. While these results were more variable than those collected in previous studies that utilized externally placed sensors, this investigation still indicates that the signals from implanted NNP sensors, which were installed without a priori knowledge of their relationship to the axes of motion of the body, have the potential to be utilized in feedback controllers to set or maintain trunk posture after SCI using FNS. Further research should explore the use of sensor fusion algorithms to create the best possible estimate of trunk tilt from inputs from multiple sensors. The results of this work will be helpful for the development of feedback control systems for the NNP that can be used regularly outside of the laboratory setting to improve performance in ADLs and the quality of life of individuals with SCI.

## Figures and Tables

**Figure 1 sensors-24-03816-f001:**
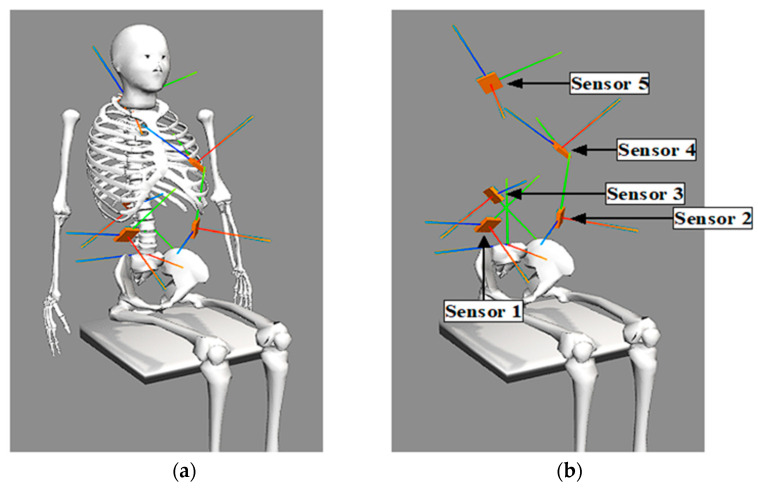
Modeled Trunk and Sensors: The modeled trunk mesh and all the sensors (**a**), as well as the sensors without the trunk and upper extremities (**b**) are depicted. Image (**b**) was included so that all sensors could be seen clearly in their respective positions.

**Figure 2 sensors-24-03816-f002:**
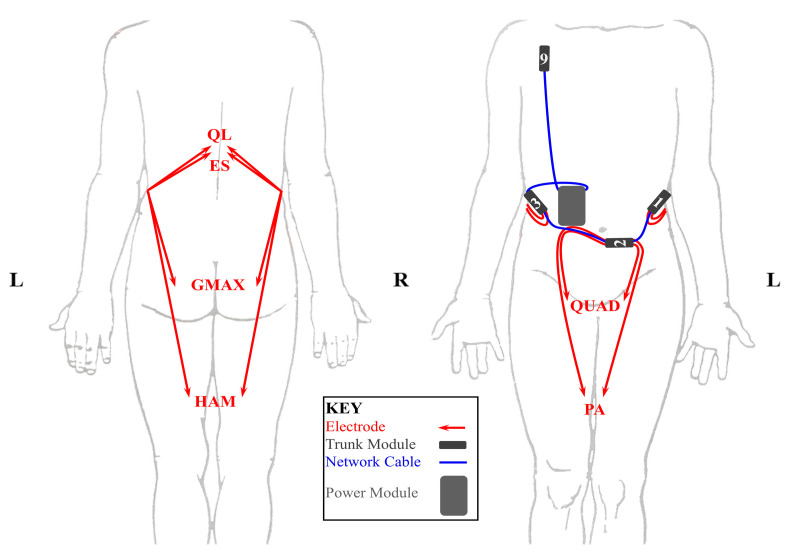
Configuration of NNP Trunk Modules: (Abbreviations: QL, quadratus lumborum; ES, erector spinae; GMAX, gluteus maximus; HAM, hamstrings; QUAD, quadriceps; PA, posterior portion of the adductor magnus).

**Figure 3 sensors-24-03816-f003:**
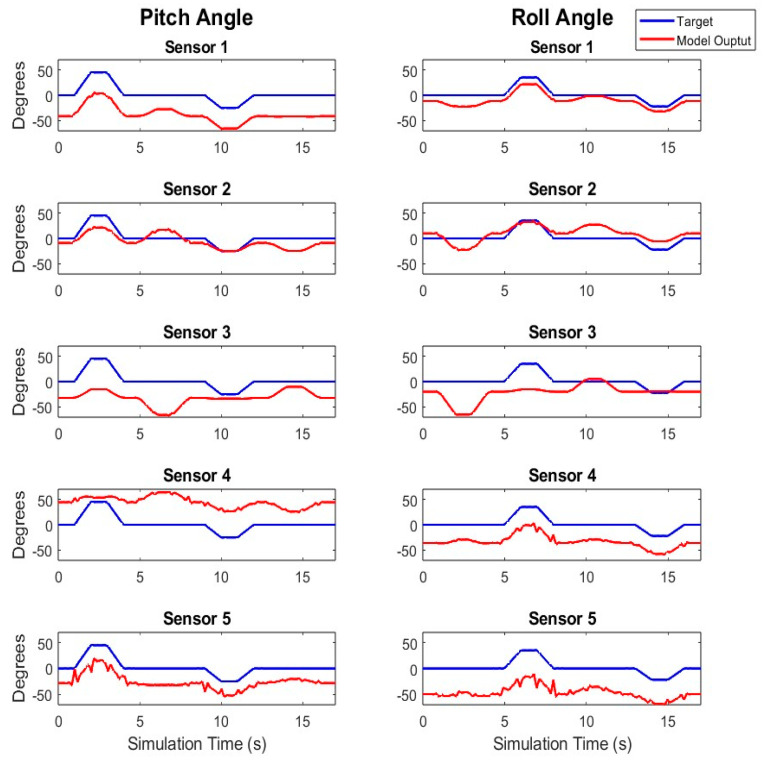
Tilt Estimation Before Sensor Reorientation: Tilt angle estimates from raw, unrotated, simulated accelerometer data with no rotation applied (red) and the expected tilt angles (blue) for the dynamic trials collected from the model.

**Figure 4 sensors-24-03816-f004:**
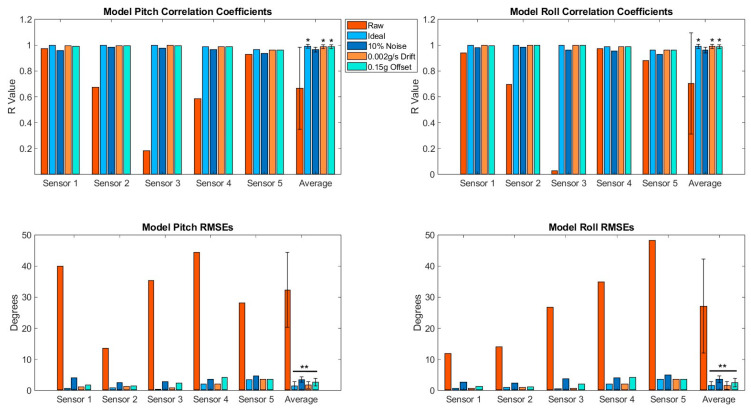
Simulated Accelerometer Signal Analysis: The averages and standard deviations of R values and RMSEs were calculated for each error measure in each condition. Asterisks indicate statistically significant differences between the unrotated and reoriented tilt estimations, and lines indicate that all groups under the line are statistically different from the raw, unrotated estimations. * *p* < 0.05, ** *p* < 0.01.

**Figure 5 sensors-24-03816-f005:**
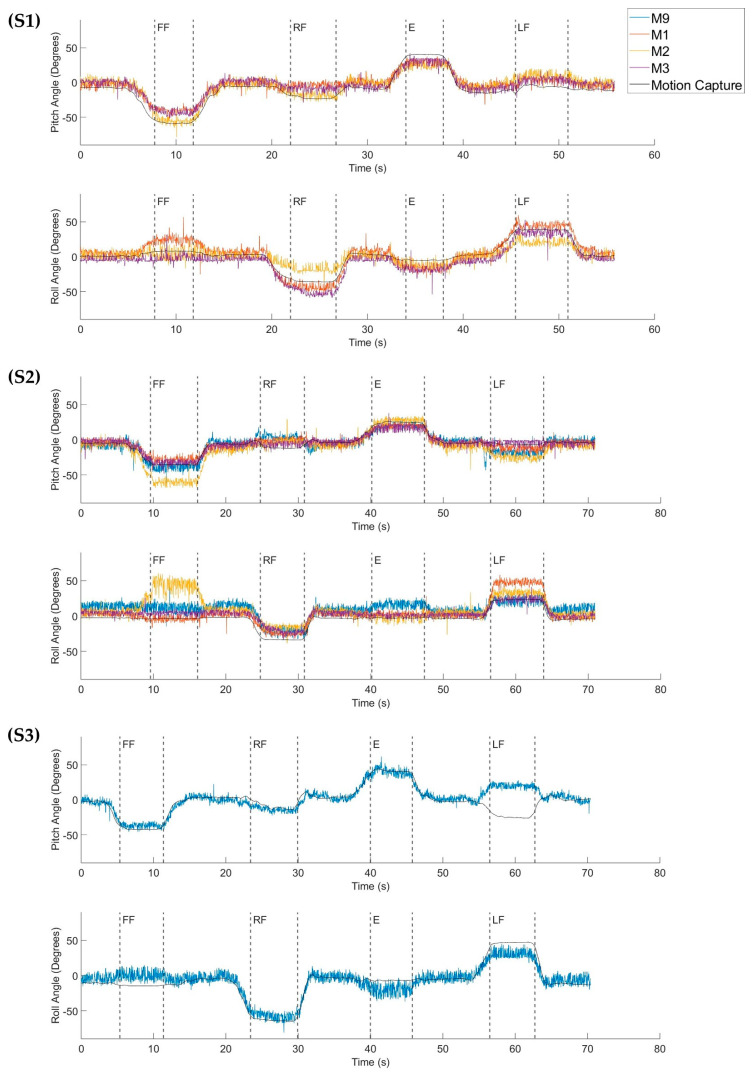
Example Tilt Estimates from Reoriented, Implanted Accelerometer Outputs: Tilt estimates collected from the accelerometers in all the available modules—M9 (module 9), M1 (module 1), M2 (module 2), and M3 (module 3)—during one of the dynamic trials for each participant (S1, S2, and S3). The motions performed are indicated by the dashed vertical lines. (Abbreviations: FF, forward flexion; RF, right flexion; E, extension; LF, left flexion).

**Figure 6 sensors-24-03816-f006:**
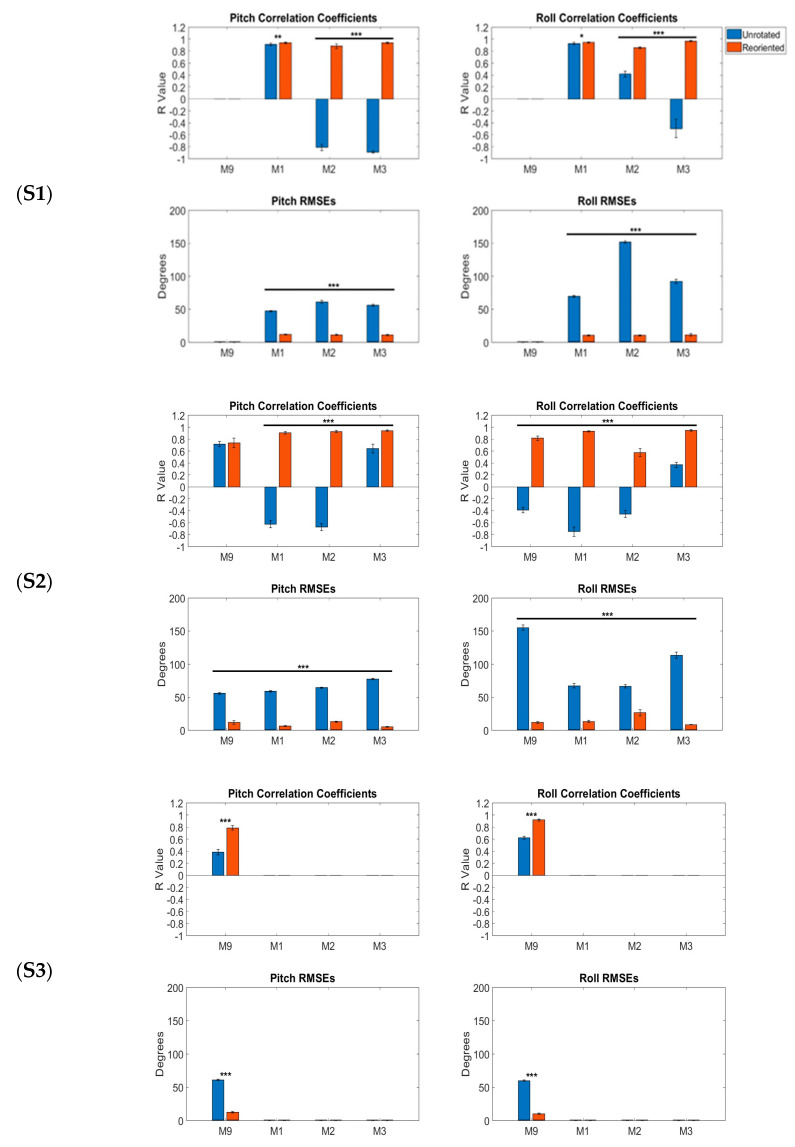
Implanted Accelerometer Signal Analysis: R values and RMSEs derived from unrotated and reoriented tilt estimates were examined for each module in each participant (S1, S2, and S3). Asterisks indicate a statistically significant difference between the unrotated and reoriented tilt estimations, and lines indicate that all groups under the line are statistically different from the unrotated estimations. * *p* < 0.05, ** *p* < 0.01, *** *p* < 0.001.

**Table 1 sensors-24-03816-t001:** Clinical data and sensor count for participants.

Subject	Gender	Age (yrs.)	Height (in)	Weight (lbs.)	Injury Level	AIS * Grade	Time Post Injury (yrs.)	Number of Modules
S1	F	58	66	89	C5–6	A	34	3
S2	M	29	72	195	C4–5	A	7	4
S3	M	53	70	170	C5	A	9	1

* American Spinal Injury Association Impact Score (AIS).

## Data Availability

The data presented in this study are available on request from the corresponding author. The data are not publicly available due to source data participant consent limitations.

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
