# Peer review of "Anatomical Registration of Implanted Sensors Improves Accuracy of Trunk Tilt Estimates with a Networked Neuroprosthesis"

_sensors, 2024, doi:10.3390/s24123816_

Round 1

Reviewer 1 Report

Comments and Suggestions for Authors

The manuscript addresses an important topic in sensor technology and neuroprosthetics. The approach of reorienting accelerometer signals using the Gram-Schmidt method shows promise in improving trunk tilt estimation for individuals with spinal cord injuries (SCI). The proposed method tested in both an in-silico study and in real-life implant recipients. Overall, the article is well-written and the results are promising. However, I have some suggestions

Comment 1: Can you explain why exactly you chose the Gram-Schmidt method for Sensor Reorientation? Are there any alternative methods?

Comment 2: Is it possible to compare the effectiveness of your proposed method with other similar studies?

Comment 3: In the introduction section, I recommend broadening the scope of the literature review to encompass a wider range of studies focusing on sensor reorientation and trunk control.  This will help guide the reader through the study, making it easier to understand the purpose and significance of your work.

Comment 4: The authors do not discuss the long-term stability of the implanted sensors. How will the sensor orientation change over time?

Some sections are heavy on technical jargon which might be challenging for readers not deeply familiar with the subject matter.

Overall Recommendation: Accept after minor revision

Comments on the Quality of English Language

It is appropriate.

Author Response

Attached as a Word document are all the responses to the comments made by the reviewer. To the reviewer, thank you for taking the time to assess this manuscript!

Reviewer 2 Report

Comments and Suggestions for Authors

The manuscript describes the application of an existing algorithm to the correction of accelerometer data without a priori knowledge of the implanted accelerometer's precise orientation and location. The demonstrations on synthetic and human data show the feasibility of the approach due to the improvement of the CC and reduction in error. However, the proposed feedback control of FNS using NNP requires real-time signal processing for accurate estimation of trunk tilt in a continuous manner. The computational time required for this proposed application is not discussed. I think some data and discussion regarding the feasibility of the proposed approach in terms of applicability to feedback control and real-time signal processing is required.

Author Response

(The authors gave the same response as above.)
